# Blackcurrants Reduce the Risk of Postmenopausal Osteoporosis: A Pilot Double-Blind, Randomized, Placebo-Controlled Clinical Trial

**DOI:** 10.3390/nu14234971

**Published:** 2022-11-23

**Authors:** Briana M. Nosal, Junichi R. Sakaki, Zachary Macdonald, Kyle Mahoney, Kijoon Kim, Matthew Madore, Staci Thornton, Thi Dong Binh Tran, George Weinstock, Elaine Choung-Hee Lee, Ock K. Chun

**Affiliations:** 1Department of Nutritional Sciences, University of Connecticut, Storrs, CT 06269, USA; 2Department of Kinesiology, University of Connecticut, Storrs, CT 06269, USA; 3Department of Food and Nutrition, Sookmyung Women’s University, Seoul 04310, Republic of Korea; 4Jackson Laboratory Center for Aging Research, Farmington, CT 06032, USA

**Keywords:** blackcurrant, osteoporosis, bone mineral density (BMD), bone remodeling, menopause, women

## Abstract

Beneficial effects of blackcurrant supplementation on bone metabolism in mice has recently been demonstrated, but no studies are available in humans. The current study aimed to examine the dose-dependent effects of blackcurrant in preventing bone loss and the underlying mechanisms of action in adult women. Forty peri- and early postmenopausal women were randomly assigned into one of three treatment groups for 6 months: (1) a placebo (control group, *n* = 13); (2) 392 mg/day of blackcurrant powder (low blackcurrant, BC, group, *n* = 16); and (3) 784 mg/day of blackcurrant powder (high BC group, *n* = 11). The significance of differences in outcome variables was tested by repeated-measures ANOVA with treatment and time as between- and within-subject factors, respectively. Overall, blackcurrant supplementation decreased the loss of whole-body bone mineral density (BMD) compared to the control group (*p* < 0.05), though the improvement of whole-body BMD remained significant only in the high BC group (*p* < 0.05). Blackcurrant supplementation also led to a significant increase in serum amino-terminal propeptide of type 1 procollagen (P1NP), a marker of bone formation (*p* < 0.05). These findings suggest that daily consumption of 784 mg of blackcurrant powder for six months mitigates the risk of postmenopausal bone loss, potentially through enhancing bone formation. Further studies of larger samples with various skeletal conditions are warranted to confirm these findings.

## 1. Introduction

Postmenopausal osteoporosis (PMO) is an estrogen deficiency-induced metabolic bone disorder, characterized by low bone mass, deterioration in bone microarchitecture, and increased bone fragility and fracture risk [1]. Osteoporosis is a major health risk for adults aged 50 years and older and its prevalence is comparable to major chronic diseases such as hypercholesterolemia (54%) and hypertension (44%) [2]. Fracture risk differs significantly between countries, possibly in part due to differences in body weight status, calcium and vitamin D intake, exposure to sunlight, smoking habits, socio-economic status, and physical activity levels [3,4]. Nonetheless, osteoporosis has become a worldwide health concern with significant medical, economic, and social burdens. 

Although hormone replacement therapy (HRT), a classical therapeutic approach, is effective in preventing the development of PMO and reducing the risk of fracture in postmenopausal women [5], the long-term acceptance of and/or compliance with HRT is low due to potential malignant effects of HRT on the reproductive tissues, as observed in the Women’s Health Initiative (WHI) Trial [6]. Thereafter, several drug classes have been developed for the prophylaxis and treatment of PMO, such as bisphosphonates, selective estrogen receptor modulators, parathyroid hormone, and denosumab; however, the safety and efficacy of long-term use of these medications are still uncertain [7,8,9,10,11,12]. Therefore, the search for novel, safe, and prophylactic dietary bioactive compounds that can improve postmenopausal bone health continues [13]. 

Natural compounds such as phytoestrogen isoflavones derived from leguminous plants and fruit and herbal extracts have been introduced as alternative therapies for the treatment of metabolic bone disorders and preservation of bone health without the deleterious side effects of currently available pharmacological treatments [14]. However, results from clinical trials with natural dietary agents have been inconsistent; while the risk of severe adverse events is minimal [15,16,17,18,19,20,21,22,23,24,25,26], only a few have demonstrated a beneficial effect on bone mineral density (BMD) in postmenopausal women [17,18,27]. Blackcurrants (*Ribes nigrum*) have drawn our special attention because they contain the greatest amount of anthocyanins among commonly consumed berries such as blueberry, blackberry, raspberry, and cranberry [28] and have been found to inhibit the formation of tartrate-resistant acid phosphatase (TRAP) (+) osteoclasts more than three-fold relative to blackberry in the presence of receptor activator of nuclear factor-κB ligand (RANKL) [29] and inhibit lipopolysaccharide (LPS)-induced inflammation [30] in in vitro studies. 

Blackcurrants effectively attenuated ovariectomy-induced [29] or aging-associated [28] bone loss in female adult mice; however, their effectiveness was sensitive to the timing of intervention initiation [31]. Peak spine and hip bone mass is reached in the mid-twenties, but other bones, such as the radius, reach a peak at age 40 in adult women. There is a rapid acceleration in bone loss that starts the year before menopause and continues for another three years [32] before de-accelerating. Even so, the rate of bone loss in the four to eight years post menopause is still high. The average decrease in BMD during the menopausal transition is about 10 percent, meaning that many women are experiencing even greater decreases, perhaps as much as 10–20 percent in the five to six years near menopause. Therefore, the initiation of dietary interventions prior to or at menopause onset rather than after bone loss has substantially progressed, and may be the best strategy for preventing PMO [31]. 

This study aimed to evaluate the dose-dependent effects of blackcurrant supplementation on bone density in adult women and its relationship to bone metabolism. The feasibility of the pilot clinical study was also assessed. For this purpose, we conducted a pilot double-blind, randomized, placebo-controlled clinical study with blackcurrant supplementation for six months in peri- and early postmenopausal women aged 45–60. The primary endpoint was whole-body BMD. In addition, to delineate the underlying mechanisms of the action, changes in serum markers of bone metabolism following blackcurrant supplementation were also assessed.

## 2. Materials and Methods

### 2.1. Study Participants

Forty peri- and postmenopausal women aged 45–60 were initially recruited from northeastern Connecticut through newspaper advertisements, flyers, and email. The inclusion criteria for this study were: (1) women aged 45–60 years old; (2) not on HRT for at least one year before the initiation of the study; (3) maintaining normal exercise level (<7 h/week) and willing to avoid exercise 24-h prior to blood and stool sampling and 12-h prior to bone measurements; (4) willing to ingest a dietary blackcurrant supplement or placebo as well as 400 mg calcium and 500 IU vitamin D daily; (5) willing to avoid other dietary supplements for the duration of the study, (6) willing to avoid intake of foods extremely rich in anthocyanins and fermented dairy products, (7) willing to have 3 blood draws and 2 bone scans and (8) willing to take a urine pregnancy test before each bone scan. Exclusion criteria included: (1) those with metabolic bone disease, renal disease, cancer, cardiovascular disease, diabetes mellitus, respiratory disease, gastrointestinal disease, liver disease, or other chronic diseases, (2) heavy smokers (>20 cigarettes/day), (3) perimenopausal women with any chance or plan of pregnancy, (4) taking prescription medications known to alter bone and Ca metabolism; (5) taking anabolic agents; (6) alcohol consumption exceeding 2 drinks/day. Subjects who met the recruitment criteria upon initial phone screening were immediately invited to the study laboratory at the University of Connecticut in Storrs, Storrs, CT, USA.

### 2.2. Study Design

Upon signing the consent form, an initial physical examination (body weight, height, waist circumference, and blood pressure) was conducted. The participants were also interviewed about their medical history and dietary behaviors by study personnel. Participants had a urine pregnancy test to confirm that they were not pregnant and thus eligible for the bone density measurement using dual-energy X-ray absorptiometry (DXA). All eligible participants completed an additional consent form for the DXA assessment and scheduled their bone mass measures on or within ±3 days of study visit 1 (month 0) and 3 (months 6) at the University of Connecticut Korey Stringer Institute Human Performance Laboratory. Participants were asked to refrain from taking the provided calcium supplements the day before the exam (24 h) to limit potential interference with the DXA measurements. 

The participants were taught how to complete 3-day food records (FR) and physical activity records and asked to complete them one week prior to each study visit at months 0, 3, and 6. Subjects who were taking any dietary supplements that are known to affect bone metabolism were asked to stop taking them. After the initial screening visit, subjects underwent a 2-week equilibration period, followed by a 6-month clinical trial period. To avoid potential bone deterioration related to calcium and vitamin D deficiency, all participants took a calcium citrate caplet daily that includes 400 mg calcium and 500 IU vitamin D (Bayer AG, Leverkusen, Germany) beginning 2 weeks before the study and lasting for the duration of the study. After the 2-week equilibration period, study participants were randomly assigned to three groups and asked to consume: (1) one capsule containing 392 mg of blackcurrant powder (low BC group); (2) two capsules containing 392 mg blackcurrant powder per capsule, total 784 mg of blackcurrant powder (high BC group); or (3) one placebo capsule (control group) daily for 6 months. Each 392 mg blackcurrant extract contained 176 mg anthocyanins (min. 40% delphinidin-3-rutinoside, 35% cyanidin-3-rutinoside, 10% delphinidin-3-glucoside and 7% cyanidin-3-glucoside). One capsule was equivalent to about 142 fresh blackcurrants. The blackcurrant powder and composition information was provided by Just the Berries, New Zealand (Just the Berries PD Corporation, Los Angeles, CA, USA). The placebo was an identical-looking capsule that contained rice powder (supplied by Beehive Botanicals Hayward, WI, USA). The extract and placebo were encapsulated in vegetarian capsules and packaged into coded containers for daily dosing to participants (Beehive Botanicals Hayward, WI). Distribution of the supplements was performed blindly at the study center at month 0 (study visit 1: after the 2-week equilibration period) and month 3 (study visit 2). 

Aside from the exclusion of dietary supplements, foods extremely rich in anthocyanins (all berries, grapes, red wine, and berry juices), and fermented dairy products containing viable *Bifidobacteria* or *Lactobacilli*, all participants were instructed to keep their usual dietary habits for the duration of the study. Subjects provided a 12-h fasting blood sample and stool specimen at each study visit at months 0, 3, and 6 for biochemical and metagenome sequencing analysis, and a physical examination (height, weight, waist circumference, blood pressure) was conducted at the initial visit (month 0), month 3, and the end of the treatment period (month 6). This clinical trial was registered at ClinicalTrials.gov (NCT04431960). The proposed project and its procedures were reviewed and approved by the University of Connecticut Institutional Review Board (HR20-0035) prior to the initiation of the project.

### 2.3. Dietary Intake and Physical Activity Assessment

During the trial, participants were asked to complete 3-day FR and physical activity records at months 0, 3, and 6. The FR data were analyzed using the Nutrition Data System for Research (NDSR, University of Minnesota Nutrition Coordinating Center, Minneapolis, MN, USA). The 3-d FR data included all foods and beverages consumed during two non-consecutive weekdays plus a weekend day of the week following the test visits. The physical activity records were used to estimate the metabolic equivalent of task (MET) scores to confirm participants maintained normal exercise levels (<7 h/week) during the entire intervention period and avoided exercise 24-h prior to study visits.

### 2.4. Bone Density Assessments

BMD analyses were conducted at the University of Connecticut Korey Stringer Institute by a licensed radiologic technician at baseline and 6 months using DXA (GE Healthcare Lunar) equipped with appropriate software for whole-body, head, arms, legs, trunk, ribs, spine, and pelvis BMD. Specific positioning and analysis guidelines were followed according to the operator’s manual. Calibration, maintenance, and quality control of the equipment were routinely conducted following the maintenance instructions in the manual. 

### 2.5. Blood Collection and Pretreatments

Fasted blood samples were collected at months 0, 3, and 6. The blood samples (80 mL) from subjects who fasted for 12 h were used to determine biomarkers for bone metabolism. Whole blood samples were collected in serum tubes (BD Vacutainer, Mississauga ON, Canada), centrifuged at 3500× *g* for 15 min at 4 °C, divided into aliquots, and stored at −80 °C until analyzed. 

### 2.6. Measurements of Serum Biomarkers of Bone Metabolism

Changes in the amino-terminal propeptide of type 1 procollagen (P1NP), bone-specific alkaline phosphatase (BALP), osteocalcin (OC), carboxy-terminal crosslinked telopeptide of type 1 collagen (CTX1), and sclerostin at baseline, 3 months, and 6 months were measured in serum using commercially available ELISA kits. 

### 2.7. Statistical Analysis

Data are reported as mean ± standard deviation (SD) unless specified otherwise. Differences with two-sided *p* < 0.05 were considered significant. Data were analyzed using analysis of variance methods with PROC MIXED in SAS (Version 9.4, SAS Institute, Cary, NC, USA) to determine the main and interaction effects of the two factors, treatment (0, 392, or 784 mg/day blackcurrants), and time (baseline, 3 months, and 6 months). The significance of differences in baseline characteristics was tested by ANOVA or Chi-square/Fisher’s Exact test. The significance of changes in all outcome variables was tested by repeated-measures ANOVA with treatment as levels of between-subjects factors and time as levels of within-subjects factors. Percent changes in BMD and bone biomarkers between control and low BC groups and between control and high BC groups were evaluated using *t*-tests. 

## 3. Results

### 3.1. Participants and Intervention Follow-Up

There were 112 individuals who responded to the advertisements for this study, 69 of whom passed the telephone pre-screening. The profile of the study participants and their treatment assignment is presented in Figure 1. After the enrollment of 54 subjects, two participants were removed from the study due to loss-to-follow-up, and an additional participant voluntarily withdrew from the study due to difficulty with compliance (excessively restrictive diet). After randomization, three participants voluntarily withdrew from the study due to non-study-related issues, such as family situations (*n* = 1) or other medical conditions (*n* = 2), and another five withdrew due to study-related issues including diet restriction (*n* = 2), failure to complete investigation drugs (*n* = 1), concerns regarding taking the calcium supplement (*n* = 1), and the need to resume other supplements (*n* = 1). Another three participants were removed from the study since they were revealed to have a bone metabolic disorder (*n* = 1) or take hormone medications (*n* = 2). Forty women (13 in Control, 16 in low BC, and 11 in high BC groups) completed the study. The attrition rate was 27.8% for Control, 5.9% for low BC, and 31.3% for high BC. Kidney function, assessed by serum creatinine levels, were normal in the study participants at baseline (Table 1) and in month three and six, indicating healthy kidney function before and after the treatment (data not shown). Additionally, there were no significant differences in the demographic characteristics of the 11 participants who did not complete the study compared to those who completed the study (data not shown). 

### 3.2. Baseline Characteristics and Intervention Compliance

Table 1 summarizes the baseline characteristics of the 40 study participants who completed the study. There were no major differences in subject characteristics between the three groups. Age, height, body weight, BMI, waist circumference, and systolic/diastolic blood pressure were similar at baseline. 

The 40 participants who remained in the study adhered to the regimens, as assessed by counting the remaining capsules and reviewing self-reported dietary supplement use records. Overall, subjects stated both doses of BC supplements (392 and 784 mg/day) were well tolerated. The compliance with the provided treatment was ≥95% for all three groups on average. Analysis of the three-day FR indicated that the participants’ nutrient intakes were not significantly different from their corresponding baseline values or between the three treatment groups throughout the study period (Table 2). Physical activity levels were also assessed at baseline, three, and six months and there were no significant differences in activity levels between the treatment groups (Table 1 and Table 3). 

### 3.3. Bone Mineral Density

Whole-body BMD of participants ranged from −0.893 (z-score of −2.4) to 1.415 (z-score of 2.5). The majority of participants (92.5%) had a normal bone mass (z-score ≥ −1), while 7.5% of participants (*n* = 3) had z-scores for whole-body BMD ranging from −1.0 to −2.5. These results are based on whole-body BMD, not at the hip and lower back, thus, cannot be used for diagnosis of osteoporosis or osteopenia. However, the data indicate that our participants are in overall good bone condition at baseline [33]. Blackcurrant supplementation for six months had bone protective effects as indicated by an increase in whole-body BMD from baseline in blackcurrant treatment groups, whereas the control group continued to lose bone (*p* < 0.05 in repeated-measures ANOVA with treatment as levels of between-subjects factors and time as levels of within-subjects factors; Table 4). When the changes (%) in whole-body BMD between the three groups were compared, the strong positive effect of blackcurrant treatments remained the same (*p* < 0.05; Figure 2A). However, when the effects of both doses were compared to the control group, only daily consumption of the high dose (784 mg) of blackcurrant significantly decreased bone loss (*p* < 0.05). The observed six-month changes from baseline for BMD of ribs in treatment groups were significantly higher compared to the control group, but this remained significant only in the high BC group (*p* < 0.05). There was a positive trend for increased pelvis BMD by blackcurrant treatment (*p* < 0.05), but the differences in both BC groups were not significantly greater than those in the control group. There was no significant difference in changes in the head, arm, leg, trunk, and spine BMD from baseline or between control, low BC, or high BC groups.

The effects of blackcurrant supplementation on the changes in BMD (%) were compared between participants who had higher bone mass (z-scores for whole-body BMD ≥ 0, *n* = 25) and those who had lower bone mass (z-scores for whole-body BMD < 0, *n* = 15) at baseline. As seen in Figure 2B,C, participants with lower BMD at baseline experienced a greater degree of increase in %BMD (+2.73% in the high BC group and +0.86% in the low BC group versus −0.97% in the control group) compared with those with higher BMD at baseline (+0.59% in the high BC group and −0.63% in the low BC group versus −1.33% in the control group), suggesting that blackcurrant supplementation might be more effective in those at greater risk of bone loss. 

### 3.4. Blood Biomarkers of Bone Metabolism and Immune-Inflammatory Status

Mean serum concentrations of biomarkers of bone metabolism, P1NP, BALP, OC, CTX1, and sclerostin were not significantly different between groups at baseline (Table 1). Blackcurrant supplementation led to a significant increase in serum P1NP, a marker of bone formation (repeated measures ANOVA with *p* < 0.05). Serum BALP, OC, CTX-1, and sclerostin were not significantly different at any time point between groups or on a time-wise basis (Table 5). 

Six-month changes in P1NP, BALP, OC, OPG, markers of bone formation, and CTX-1 and sclerostin, markers of bone resorption, as well as BALP/CTX1 ratio, are shown in Figure 3. Although the trends were non-significant, there was an increase in bone formation markers and bone formation/resorption index (BALP/CTX1) at six months following blackcurrant supplementation. 

## 4. Discussion

All three groups in the present study had similar baseline characteristics at the start of the study, confirming adequate randomization. Further, the average compliance rates of ≥95% for all three groups are excellent. Likewise, the overall attrition rate of 21.6% for a six-month clinical trial which was conducted in the middle of the COVID-19 pandemic is reasonable. 

We previously showed that blackcurrants lowered inflammation and osteoclastogenesis in in vitro studies and mitigated estrogen deficiency-induced [29] or aging-associated [31] bone loss in female adult mice. These findings led us to hypothesize that blackcurrants could exert beneficial effects on estrogen deficiency-induced bone loss in adult women via reducing bone resorption. The current study confirmed this hypothesis as evidenced by the significant increase of whole-body BMD after a daily supplementation of 784 mg blackcurrant for six months; however, the bone protective effects might be through enhancing bone formation as well as suppressing bone resorption as demonstrated by the tendency of bone formation markers and bone formation/resorption index (BALP/CTX1) to increase with blackcurrant supplementation. Unlike our previous animal studies and the majority of human clinical trials for the prevention of PMO in postmenopausal women, the current study targeted peri- and early postmenopausal women who are relatively young (average age of 53.1 ± 4.3 years) and retained sufficient bone mass (BMD Z-score ranged from −2.4 to 2.5 with 7.5% increased risk of osteopenia but no osteoporosis cases at baseline) [33]. At this age, bone formation still plays a significant role in bone remodeling through interplay with bone resorption. Recent research has identified delphinidin-3-rutinoside, a major anthocyanin (44%) in blackcurrant [30], affects osteoblast differentiation by activating the fibroblast growth factor (FGF) pathway [34], demonstrating its potential as a bone stimulating agent for the prevention of PMO.

Several studies have documented the bone protective effects of antioxidant-rich fruits such as blueberries [35] and dried plums [36,37,38], but the effects of blackcurrants on bone health and their mechanisms of action have remained unexplored. Furthermore, although a number of clinical trials have been conducted to prove the bone-protective effects of non-pharmaceutical natural agents, only a few have successfully demonstrated a beneficial effect on BMD in postmenopausal women [17,18,27]. To our knowledge, this is the first randomized clinical trial that investigated how blackcurrant supplementation could mitigate estrogen deficiency-induced bone loss in adult women. To maximize the treatment effects and benefits of the participants, our clinical trial targeted early intervention by enrolling peri-and early postmenopausal women aged 45–60 years. 

Several research groups have conducted randomized clinical trials investigating the effect of plums on bone metabolism [17,18,27,39,40,41]. However, changes in bone metabolism biomarkers by plum treatments were not consistent: for example, three months of dried plum consumption significantly increased levels of BALP in postmenopausal women [17], but decreased [40] or produced no significant changes in BALP in men after three [41] or six months [27]. Similarly, results regarding CTX1, OC, or other bone markers were inconsistent in these studies. The authors discussed that such discrepancies might be attributable to gender differences, lifestyle characteristics such as alcohol drinking, or a combination of these factors that might modify responsiveness to the intervention [41]. It was also suggested that blood biomarkers might not be fully indicative of what is happening at the cellular level and should be interpreted with caution, and that greater value should be placed on the resulting BMD values as they are more clinically relevant [27]. In addition, it is noteworthy that the above-mentioned studies targeted postmenopausal women and men with some degree of bone loss (e.g., osteopenia), whereas our study included peri- and early postmenopausal women aged 45–60 years who likely have important differences in bone physiology. Thus, it must be emphasized that the findings from this study are relevant to general adult women in the menopause transition.

In the current study, six months of blackcurrant supplementation led to a significant increase in serum P1NP, especially in the high-dose group. Furthermore, although the trends were non-significant, blackcurrant supplementation showed a tendency to mitigate decreases in BALP, OC, and the BALP/CTX1 ratio while reducing sclerostin, an important mechanosensor that inhibits osteoblastic bone formation [34]. We also found that the bone protective effects of blackcurrant supplementation might be different by the degree of progression of postmenopausal bone loss. Blackcurrant supplementation might be more effective in those at greater risk of bone loss. Due to small sample sizes in the subgroup analysis, these trends were not statistically significant, warranting further investigation in studies with a larger number of participants and various levels of bone health status. Our results imply that blackcurrants may reduce the risk of PMO, and, therefore, could serve as a preventive agent in women who still retain adult bone mass or as a therapeutic agent in women whose bone loss has substantially progressed.

Our trial has several strengths. First, adherence to study intervention was excellent, attrition rates were reasonable, and participants kept their usual diets and physical activity levels throughout the entire study. Second, this is one of few intervention studies that targeted women in menopause transition rather than postmenopausal women. Thus, the findings of our study indicate the potential for blackcurrant supplementation to serve as a prophylactic against PMO. Lastly, this is the first human clinical study that investigated whether blackcurrant supplementation could mitigate estrogen deficiency-induced bone loss and sought to identify mechanisms of action.

Some limitations of the current study include the sample size, as a larger sample size would achieve greater power to observe significant changes in serum inflammation and bone markers. Second, a six-month intervention may not be enough time to fully observe intervention-induced effects on bone density. Perhaps, longer exposure to blackcurrants, through more bone remodeling cycles, would yield greater effects on BMD. Third, although randomly recruited, our participants were not ethnically diverse, with most of the sample being Caucasian. Fourth, while the study was double-blind, participants were not blinded to taking one versus two capsules. Those taking two capsules might have been able to infer that they were not taking the placebo, although they did not know how many capsules others were taking. Lastly, estrogen levels were not monitored through the course of the study, which is known to be associated with bone metabolism in women and may have had an influence on bone health in this population.

## 5. Conclusions

The current study demonstrated the bone protective effects of a 784 mg blackcurrant supplementation in women in the menopause transition. We focused on an early intervention since we previously found that this strategy was the most effective. These promising results warrant further investigation, especially using samples with a broader age range, including women aged 65 years or older, which allows us to confirm the relative effectiveness of blackcurrant supplementation in multiple adult female subpopulations. Numerous studies employing various methods have dealt with the pathophysiology of PMO, osteoporosis type 1 caused by estrogen deficiency in women in menopause transition [1]. In contrast, age-associated osteoporosis (AAO), osteoporosis type 2, which is also called senile osteoporosis and related to bone mass loss due to aging, has not been well studied [42]. Thus, a larger study on an adult population with a broader age range would expand the body of knowledge upon which dietary recommendations for this growing senior population could be established. 

In this double-blind randomized clinical trial with peri- and early postmenopausal women, daily supplementation of 784 mg of blackcurrant extract over six months was effective in preventing bone loss. Whether a lower dose (i.e., 392 mg/day) may be as effective as 784 mg of blackcurrant needs confirmation via additional studies with larger numbers of participants. Furthermore, longer-term studies are also necessary to confirm the findings of this study, as bone-building metabolism is a lengthy process, typically taking years. Thus, conducting a trial one year or longer with a larger sample size may reveal significant differences in sites such as the pelvis and ribs as well as greater improvements in the whole-body BMD. Overall, the findings from this pilot randomized clinical trial on bone health are promising and confirm the bone-protective effects of blackcurrants in adult women in the menopause transition.

## Figures and Tables

**Figure 1 nutrients-14-04971-f001:**
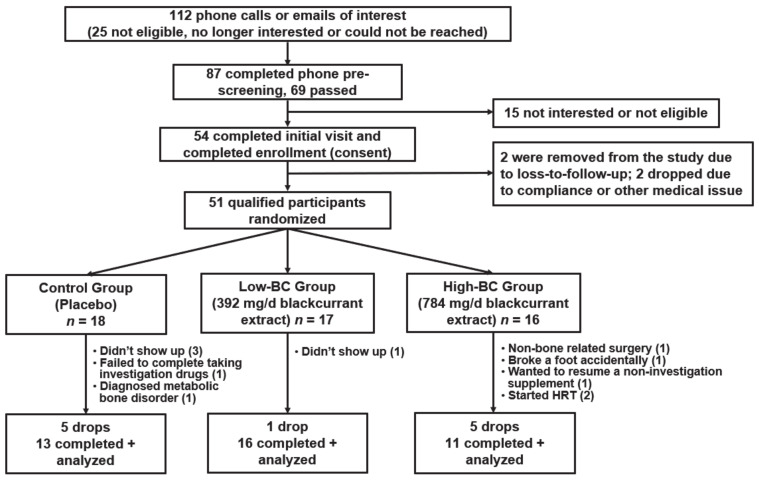
Inclusion and follow-up of the study participants. Final analysis *n* = 40.

**Figure 2 nutrients-14-04971-f002:**
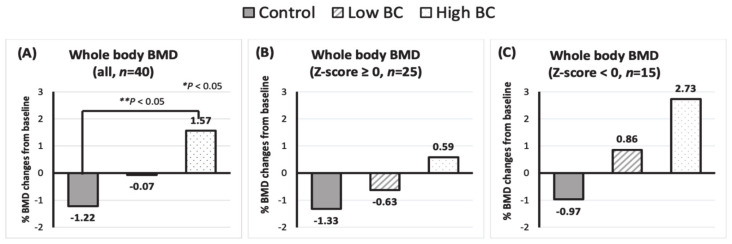
Changes (%) in bone mineral density (BMD) of whole body from baseline to 6 months of consumption of placebo (Control), 392 mg (low BC) and 784 mg (high BC) of blackcurrant powder for (**A**) all participants, (**B**) participants with z-scores ≥ 0, and (**C**) participants with z-score < 0. Bars represent mean. * Significance was assessed using one-way ANOVA. ** Significance was assessed by *t*-test for control and low BC and control and high BC groups, respectively.

**Figure 3 nutrients-14-04971-f003:**
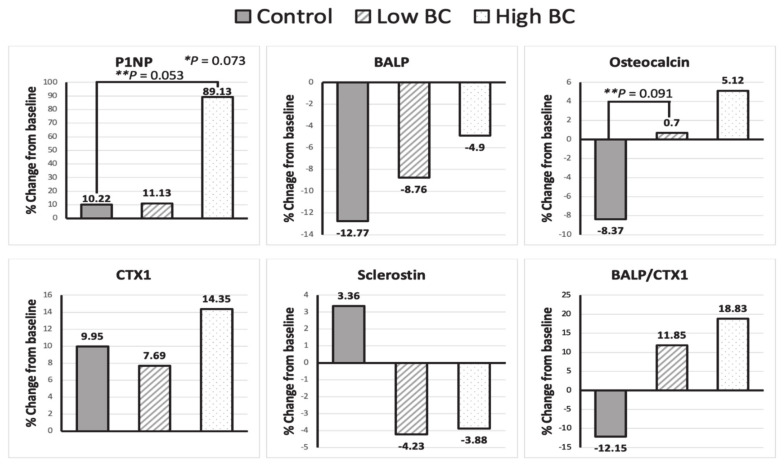
Changes (%) of amino-terminal propeptide of type 1 procollagen (P1NP), bone-specific alkaline phosphatase (BALP), osteocalcin (OC), carboxy-terminal crosslinked telopeptide of type 1 collagen (CTX-1), sclerostin, and BALP/CTX1 ratio from baseline after six months of consumption of placebo (control), 392 (low BC) and 784 mg (high BC) of blackcurrant powder. * Significance was assessed using one-way ANOVA. ** Significance was assessed using t-tests for comparing control and low BC and control and high BC groups, respectively.

**Table 1 nutrients-14-04971-t001:** Baseline characteristics (mean, SD) of study participants ^a^.

	Control	Low BC	High BC	*p*-Value ^b^
	(Placebo, *n* = 13)	(392 mg/Day, *n* = 16)	(784 mg/Day, *n* = 11)	(ANOVA/Chi-Sq or Fisher)
Age (year)	54.4 ± 3.8	53.7 ± 4.5	50.9 ± 4.2	0.128
Weight (kg)	73.2 ± 17.7	72.4 ± 15.2	71.4 ± 14.6	0.965
Height (cm)	164.9 ± 4.9	164.0 ± 6.5	166.1 ± 6.1	0.658
BMI (kg/m^2^)	27.0 ± 7.2	26.8 ± 4.9	25.8 ± 4.9	0.866
Waist circumference (cm)	85.5 ± 15.6	85.2 ± 11.6	81.8 ± 12.0	0.747
Physical activity (MET-min/day)	457.5 ± 350.5	472.7 ± 353.2	386.4 ± 309.4	0.814
Systolic blood pressure (mm Hg)	115.7 ± 14.1	118.8 ± 15.0	109.9 ± 13.5	0.318
Diastolic blood pressure (mm Hg)	80.5 ± 7.1	81.0 ± 7.5	76.5 ± 7.4	0.274
Race/ethnicity (n, %)				0.373
Caucasian	13, 100%	13, 81.3%	11, 100%	
Hispanic	0	1, 6.3%	0	
Asian-American/Pacific Islander	0	1, 6.3%	0	
African American	0	0	0	
Native American	0	0	0	
Other	0	0	0	
Marital status (n, %)				0.104
Single	0	0	2, 18.2%	
Married	10, 76.9%	15, 93.8%	7, 63.6%	
Divorced	3, 23.1%	1, 6.3%	1, 9.1%	
Widowed	0	0	1, 9.1%	
Other	0	0	0	
Annual household income				0.583
<$20,000	0	0	0	
$20,000–$45,000	1, 7.7%	0	0	
$45,000–$65,000	2, 15.4%	0	0	
$65,000–$90,000	2, 15.4%	1, 6.3%	2, 18.2%	
$90,000–$125,000	3, 23.1%	6, 37.5%	4, 36.4%	
>$125,000	5, 38.5%	9, 56.3%	5, 45.5%	
Highest educational attainment (n, %)			0.398	
Attended high school	0	0	0	
Graduated high school	1, 7.7%	1, 6.3	1, 9.1%	
Attended college	3, 23.1%	1, 6.3%	1, 9.1%	
Undergraduate degree	6, 46.1%	5, 31.3%	2, 18.2%	
Graduate degree	3, 23.1%	9, 56.3%	7, 63.6%	
Menopause (n, %)	8, 61.5%	13, 81.3%	7, 63.6%	0.058
Energy intake (kcal/day)	1539.0 ± 481.4	1638.8 ± 550.1	1460.7 ± 375.4	0.640
Serum creatinine (mg/dL) ^c^	0.84 ± 0.10	0.82 ± 0.12	0.80 ± 0.15	0.663
Bone mineral density, BMD				
Whole body BMD (g/cm^2^)	1.17 ± 0.13	1.14 ± 0.13	1.15 ± 0.08	0.738
Head BMD (g/cm^2^)	2.30 ± 0.32	2.30 ± 0.35	2.40 ± 0.22	0.644
Arms BMD (g/cm^2^)	0.88 ± 0.12	0.85 ± 0.13	0.80 ± 0.12	0.265
Legs BMD (g/cm^2^)	1.20 ± 0.15	1.13 ± 0.11	1.14 ± 0.06	0.265
Trunk BMD (g/cm^2^)	0.95 ± 0.12	0.93 ± 0.13	0.96 ± 0.08	0.863
Ribs BMD (g/cm^2^)	0.75 ± 0.09	0.76 ± 0.11	0.78 ± 0.14	0.847
Spine BMD (g/cm^2^)	1.11 ± 0.16	1.10 ± 0.21	1.12 ± 0.11	0.947
Pelvis BMD (g/cm^2^)	1.04 ± 0.13	1.00 ± 0.13	1.03 ± 0.13	0.712
Biomarkers of bone metabolism				
P1NP (ng/mL)	32.53 ± 20.93	26.55 ± 10.43	21.21 ± 10.09	0.184
BALP (U/L)	17.52 ± 6.18	19.43 ± 8.77	17.57 ± 7.30	0.753
OC (ng/mL)	10.05 ± 3.96	9.07 ± 3.66	9.04 ± 2.00	0.693
CTX1 (ng/mL)	1.11 ± 0.59	0.96 ± 1.02	1.13 ± 0.85	0.842
Sclerostin (ng/mL)	0.72 ± 0.22	0.70 ± 0.22	0.75 ± 0.33	0.863

P1NP, amino-terminal propeptide of type 1 procollagen; BALP, bone-specific alkaline phosphatase; OC, osteocalcin; CTX1, for carboxy-terminal crosslinked telopeptide of type 1 collagen. ^a^ Data are mean (standard deviation) unless specified otherwise. ^b^ Significance of differences in baseline characteristics were tested by ANOVA or Chi-square/Fisher’s Exact test. ^c^ None were above the reference range: females 0.6 to 1.1 mg/dL (53 to 97.2 µmol/L).

**Table 2 nutrients-14-04971-t002:** Daily average nutrient intake of study participants calculated from three-day food records at baseline, three months, and six months.

		Control	Low BC	High BC	*p*-Value *
		(Placebo, *n* = 13)	(392 mg/Day, *n* = 16)	(784 mg/Day, *n* = 11)	
Energy (kcal)	baseline	1539.0 ± 481.4	1638.8 ± 550.1	1460.7 ± 375.4	0.201
	3 months	1538.5 ± 532.2	1658.7 ± 481.1	1509.2 ± 449.6	
	6 months	1678.5 ± 553.2	1554.6 ± 490.5	1728.9 ± 524.3	
Fat (g)	baseline	67.0 ± 31.0	66.7 ± 28.1	64.4 ± 29.7	0.468
	3 months	70.8 ± 34.0	67.9 ± 28.7	62.3 ± 25.7	
	6 months	72.0 ± 32.4	65.6 ± 34.4	79.4 ± 47.3	
Carbohydrate (g)	baseline	171.9 ± 42.5	194.7 ± 64.7	154.7 ± 38.2	0.162
	3 months	162.6 ± 60.3	201.3 ± 63.5	169.3 ± 59.4	
	6 months	183.7 ± 76.9	181.1 ± 49.3	183.3 ± 60.7	
Fiber (g)	baseline	17.6 ± 4.2	20.9 ± 8.3	18.8 ± 4.9	0.375
	3 months	16.8 ± 4.9	22.0 ± 7.7	22.4 ± 8.2	
	6 months	16.7 ± 7.4	22.2 ± 8.4	22.9 ± 8.2	
Protein (g)	baseline	60.3 ± 14.4	66.3 ± 22.6	59.6 ± 16.2	0.421
	3 months	62.8 ± 24.7	68.3 ± 22.1	66.6 ± 19.3	
	6 months	70.8 ± 16.3	64.6 ± 26.6	71.4 ± 18.1	
Saturated fatty acids (g)	baseline	26.5 ± 14.4	23.4 ± 13.0	21.4 ± 10.9	0.599
	3 months	27.2 ± 14.2	24.0 ± 12.2	20.6 ± 9.0	
	6 months	29.1 ± 16.4	22.0 ± 12.7	25.3 ± 14.8	
Trans-fatty acids (g)	baseline	1.8 ± 1.4	1.7 ± 2.0	1.4 ± 0.7	0.913
	3 months	1.7 ± 1.3	1.7 ± 1.7	1.1 ± 0.6	
	6 months	1.7 ± 1.4	1.4 ± 0.9	1.4 ± 0.9	
Linoleic acids (g)	baseline	9.2 ± 4.9	11.6 ± 6.6	12.6 ± 8.2	0.246
	3 months	10.8 ± 6.7	11.2 ± 4.8	12.4 ± 7.4	
	6 months	9.7 ± 4.0	11.1 ± 6.0	17.8 ± 15.0	
Alpha-linolenic acids (g)	baseline	0.9 ± 0.4	1.4 ± 1.0	1.6 ± 1.0	0.245
	3 months	1.3 ± 0.7	1.1 ± 0.5	1.7 ± 1.3	
	6 months	1.2 ± 0.5	1.1 ± 0.5	2.1 ± 1.9	
Vitamin A (µg RAE)	baseline	610.0 ± 255.8	790.8 ± 507.3	842.3 ± 349.8	0.585
	3 months	607.8 ± 158.0	860.1 ± 424.7	791.8 ± 654.5	
	6 months	850.9 ± 493.9	1019.3 ± 805.2	734.4 ± 320.8	
Vitamin C (mg)	baseline	66.0 ± 40.3	86.7 ± 51.4	72.7 ± 40.7	0.788
	3 months	58.5 ± 34.3	78.0 ± 49.7	67.0 ± 42.7	
	6 months	60.6 ± 47.6	74.5 ± 59.7	83.6 ± 35.8	
Vitamin D (µg)	baseline	3.4 ± 2.2	4.0 ± 3.1	3.4 ± 2.7	0.147
	3 months	3.7 ± 2.8	3.9 ± 2.6	2.9 ± 1.8	
	6 months	6.0 ± 6.1	3.4 ± 2.9	5.8 ± 4.5	
Vitamin E (mg α-tocopherol)	baseline	8.4 ± 5.8	8.9 ± 4.2	8.9 ± 3.5	0.397
	3 months	6.9 ± 3.8	9.9 ± 5.7	8.0 ± 4.5	
	6 months	7.1 ± 2.8	10.6 ± 10.1	10.9 ± 4.8	
Vitamin K (µg)	baseline	86.7 ± 52.9	224.0 ± 418.1	122.4 ± 57.7	0.751
	3 months	106.0 ± 81.6	174.9 ± 180.6	178.1 ± 123.0	
	6 months	100.8 ± 89.5	191.2 ± 236.3	189.0 ± 129.4	
Calcium (mg)	baseline	630.1 ± 294.1	794.1 ± 355.3	684.0 ± 265.2	0.385
	3 months	696.2 ± 238.9	808.5 ± 290.7	736.3 ± 331.0	
	6 months	738.5 ± 200.1	698.9 ± 291.6	713.3 ± 305.3	
Iron (mg)	baseline	10.8 ± 3.6	13.5 ± 4.9	11.7 ± 4.7	0.222
	3 months	10.8 ± 3.3	16.0 ± 6.4	13.7 ± 6.1	
	6 months	11.7 ± 3.7	13.6 ± 4.7	14.8 ± 4.2	
Magnesium (mg)	baseline	256.5 ± 76.0	283.9 ± 110.5	256.7 ± 68.5	0.348
	3 months	232.3 ± 58.0	313.0 ± 115.5	314.5 ± 149.1	
	6 months	261.6 ± 87.9	306.1 ± 142.1	320.2 ± 104.8	
Phosphorus (mg)	baseline	922.8 ± 243.9	1013.1 ± 334.6	925.0 ± 216.5	0.150
	3 months	957.1 ± 272.6	1126.6 ± 336.4	1128.7 ± 358.6	
	6 months	1091.3 ± 218.2	1016.2 ± 359.2	1120.3 ± 351.6	

* Significance of differences were assessed using repeated-measures ANOVA with treatment as levels of between-subjects factors and time as levels of within-subjects factors. Post-hoc tukey test for significance was performed if ANOVA was <0.05.

**Table 3 nutrients-14-04971-t003:** Anthropometric characteristics, blood pressure and physical activity (mean, SD) of study participants at baseline, three months, and six months.

		Control	Low BC	High BC	
		(Placebo, *n* = 13)	(392 mg/Day, *n* = 16)	(784 mg/Day, *n* = 11)	*p*-Value *
Weight (kg)	baseline	73.2 ± 17.7	72.4 ± 15.2	71.4 ± 14.6	0.779
	3 months	74.1 ± 18.2	73.0 ± 15.6	72.1 ± 15.1	
	6 months	73.9 ± 18.0	72.9 ± 15.4	71.2 ± 14.6	
BMI (kg/m^2^)	baseline	27.0 ± 7.2	26.8 ± 4.9	25.8 ± 4.9	0.746
	3 months	27.4 ± 7.5	27.0 ± 5.0	25.8 ± 5.0	
	6 months	27.3 ± 7.5	27.0 ± 5.0	25.8 ± 4.9	
WC (cm)	baseline	85.5 ± 15.6	85.2 ± 11.6	81.8 ± 12.0	0.345
	3 months	86.1 ± 15.4	87.9 ± 11.8	83.4 ± 13.0	
	6 months	86.6 ± 15.4	88.6 ± 12.9	83.5 ± 11.0	
SBP (mmHg)	baseline	115.7 ± 14.1	118.5 ± 15.0	109.9 ± 13.5	0.192
	3 months	115.6 ± 14.3	111.8 ± 14.3	113.3 ± 10.8	
	6 months	115.2 ± 11.9	113.5 ± 13.7	112.6 ± 11.5	
DBP (mmHg)	baseline	80.5± 7.1	81.0 ± 7.5	76.5 ± 7.5	0.865
	3 months	79.9 ± 10.5	79.2 ± 9.9	77.6 ± 7.2	
	6 months	79.6 ± 12.2	80.8 ± 9.7	76.8 ± 6.3	
PA (MET-min/day)	baseline	457.5 ± 350.5	472.7 ± 353.2	386.4 ± 309.4	0.157
	3 months	353.8 ± 278.1	552.8 ± 388.9	444.3 ± 464.3	
	6 months	372.4 ± 306.7	647.2 ± 410.1	471.6 ± 325.9	

WC stands for waist circumference; SBP, systolic blood pressure; DBP, diastolic blood pressure; PA, physical activity. * Significance of differences were assessed using repeated-measures ANOVA with treatment as levels of between-subjects factors and time as levels of within-subjects factors. Post-hoc tukey test for significance was performed if ANOVA was <0.05.

**Table 4 nutrients-14-04971-t004:** Bone mineral density (BMD, g/cm^2^) of whole-body, head, arms, legs, trunk, ribs, spine, and pelvis at baseline and after 6 months of placebo (Control), 392 (low BC) and 784 mg/day (high BC) of blackcurrant powder treatment.

BMD		Control	Low BC	High BC			
(g/cm^2^)		(Placebo, *n* = 13)	(392 mg/Day, *n* = 16)	(784 mg/Day, *n* = 11)	*p*-Value *	*p*-Value **	*p*-Value ***
Whole-body BMD	baseline	1.17 ± 0.13	1.14 ± 0.13	1.15 ± 0.08	<0.05	0.214	<0.05
	6 months	1.16 ± 0.13	1.14 ± 0.12	1.17 ± 0.08			
Head-BMD	baseline	2.30 ± 0.32	2.30 ± 0.35	2.40 ± 0.22	0.965	0.889	0.886
	6 months	2.31 ± 0.32	2.29 ± 0.34	2.40 ± 0.23			
Arms-BMD	baseline	0.88 ± 0.12	0.85 ± 0.13	0.80 ± 0.12	0.081	0.969	0.080
	6 months	0.87 ± 0.13	0.84 ± 0.13	0.86 ± 0.06			
Legs-BMD	baseline	1.20 ± 0.15	1.13 ± 0.11	1.14 ± 0.06	0.355	0.276	0.225
	6 months	1.18 ± 0.13	1.13 ± 0.10	1.14 ± 0.08			
Trunk-BMD	baseline	0.95 ± 0.12	0.93 ± 0.13	0.96 ± 0.08	0.241	0.289	0.065
	6 months	0.94 ± 0.12	0.94 ± 0.14	0.97 ± 0.09			
Ribs-BMD	baseline	0.75 ± 0.09	0.76 ± 0.11	0.78 ± 0.14	<0.05	0.389	<0.05
	6 months	0.74 ± 0.07	0.76 ± 0.11	0.81 ± 0.14			
Spine-BMD	baseline	1.11 ± 0.16	1.10 ± 0.21	1.12 ± 0.11	0.737	0.672	0.735
	6 months	1.11 ± 0.17	1.08 ± 0.22	1.12 ± 0.12			
Pelvis-BMD	baseline	1.04 ± 0.13	1.00 ± 0.13	1.03 ± 0.13	<0.05	0.137	0.922
	6 months	1.03 ± 0.12	1.02 ± 0.15	1.02 ± 0.12			

* Significance of differences were assessed using repeated-measures ANOVA with treatment as levels of between-subjects factors and time as levels of within-subjects factors. ** Significance of differences were assessed using repeated-measures ANOVA only for control and low BC groups. *** Significance of differences were assessed using repeated-measures ANOVA only for control and high BC groups.

**Table 5 nutrients-14-04971-t005:** Effect of different doses of blackcurrant powder on biomarkers of bone metabolism and immune-inflammatory status at baseline, three months, and six months.

		Control	Low BC	High BC			
		(Placebo, *n* = 13)	(392 mg/Day, *n* = 16)	(784 mg/Day, *n* = 11)	*p*-Value *	*p*-Value **	*p*-Value ***
Biomarkers of bone metabolism							
P1NP (ng/mL)	baseline	32.53 ± 20.93	26.55 ± 10.43	21.21 ± 10.09	<0.05	0.400	<0.05
	3 months	27.19 ± 9.79	26.78 ± 10.82	25.86 ± 15.00			
	6 months	26.36 ± 17.71	25.99 ± 10.27	41.79 ± 39.47			
BALP (U/L)	baseline	17.52 ± 6.18	19.43 ± 8.77	17.57 ± 7.30	0.679	0.256	0.533
	3 months	14.87 ± 5.30	18.42 ± 9.58	16.95 ± 7.14			
	6 months	14.81 ± 5.11	17.29 ± 8.59	16.11 ± 6.91			
OC (ng/mL)	baseline	10.05 ± 3.96	9.07 ± 3.66	9.04 ± 2.00	0.269	0.375	0.109
	3 months	9.75 ± 3.69	9.35 ± 4.32	9.53 ± 2.30			
	6 months	8.84 ± 2.78	9.15 ± 3.81	9.57 ± 3.61			
CTX1 (ng/mL)	baseline	1.11 ± 0.59	0.96 ± 1.02	1.13 ± 0.85	0.475	0.428	0.881
	3 months	1.26 ± 1.12	0.84 ± 0.68	0.77 ± 0.72			
	6 months	1.36 ± 1.14	0.93 ± 0.81	1.31 ± 1.56			
Sclerostin (ng/mL)	baseline	0.72 ± 0.22	0.70 ± 0.22	0.75 ± 0.33	0.803	0.816	0.265
	3 months	0.69 ± 0.21	0.65 ± 0.17	0.69 ± 0.24			
	6 months	0.73 ± 0.19	0.64 ± 0.16	0.69 ± 0.28			
BALP/CTX1	baseline	23.52 ± 21.90	44.93 ± 45.25	32.32 ± 37.26	0.713	0.561	0.129
	3 months	20.96 ± 22.95	47.56 ± 60.49	39.66 ± 48.25			
	6 months	21.04 ± 22.21	52.39 ± 55.80	42.59 ± 61.70			

P1NP, amino-terminal propeptide of type 1 procollagen; BALP, bone-specific alkaline phosphatase; OC, osteocalcin; CTX-1, for carboxy-terminal crosslinked telopeptide of type 1 collagen. * Significance of differences were assessed using repeated-measures ANOVA with treatment as levels of between-subjects factors and time as levels of within-subjects factors. Post-hoc tukey test for significance was performed if ANOVA was <0.05. ** Significance was assessed using repeated-measures ANOVA only for control and low BC groups. *** Significance was assessed using repeated-measures ANOVA only for control and high BC groups.

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
