# Peer review of "Blackcurrants Reduce the Risk of Postmenopausal Osteoporosis: A Pilot Double-Blind, Randomized, Placebo-Controlled Clinical Trial"

_nutrients, 2022, doi:10.3390/nu14234971_

Round 1

Reviewer 1 Report

In this manuscript, the authors demonstrated the bone protective effects of blackcurrant supplementation in peri- and postmenopausal women. The manuscript is well written in general. I only have a few comments/suggestions for improvements as follows.

1.      Abstract & Introduction – The abbreviations need to be spelled out the first time using it.

2.      Method – how was the dose of blackcurrant determined?

3.      Inclusion/Exclusion – was there a cut-off for habitual calcium intake level used when enrolling participants?

4.      Did the authors consider measuring serum 25D levels?

5.      Figure 3 – the way the data is presented is very confusing and misleading. I am assuming that no statistical significance may be due to the big variation, but the figure shows obvious significant changes.

6.      Table 5 – what is the possible reason behind the big variation between groups at baseline levels, especially for RANKL?

7.      Line 527 – “only one research team” is inaccurate. In fact, there is a recently published paper on prunes: De Souza et al., Prunes preserve hip bone mineral density in a 12-month randomized controlled trial in postmenopausal women: the Prune Study. The American Journal of Clinical Nutrition, nqac189, https://doi.org/10.1093/ajcn/nqac189

Reviewer 2 Report

"Blackcurrants shape distinct gut microbiota profile and reduce the risk of postmenopausal osteoporosis: A pilot double-blind, randomised, placebo-controlled clinical trial" by Nosal and colleagues is an interesting manuscript investigating the dose-dependent effects of blackcurrants in preventing bone loss and the underlying mechanisms of action in thirty-six peri- and early postmenopausal women. The authors demonstrated that whole-body BMD loss was significantly lower with both doses of blackcurrants than with the control group. Furthermore, changes in whole-body BMD over 6 months were inversely correlated with those of CTX-1 and other immune-inflammatory biomarkers, such as IL-4, IL-6, TNF and IL-10. In addition, the authors suggest that daily consumption of 784 mg blackcurrant for 6 months mitigates the risk of postmenopausal bone loss potentially through improved gut microbial balance.

Overall, the manuscript is well structured and the results are innovative and interesting. I have only a few minor revisions to recommend to the authors. First of all, I suggest emphasising in the introduction how widespread osteoporosis is, being a public health problem with rather high socio-economic and health costs. In this regard, a review was recently published that addresses the topic well (Tarantino U, Cariati I, Greggi C, Iundusi R, Gasbarra E, Iolascon G, Kurth A, Akesson KE, Bouxsein M, Tranquilli Leali P, Civinini R, Falez F, Brandi ML. Gaps and alternative surgical and non-surgical approaches in the bone fragility management: an updated review. Osteoporos Int. 2022 Jul 18. doi: 10.1007/s00198-022-06482-z. Epub ahead of print. PMID: 35851407.).

Secondly, the authors should indicate in Figures 2 and 3 the standard error and statistical significance for each graph shown.

Finally, I suggest the authors review the English language extensively.

These corrections will certainly improve the quality of the manuscript. 
